# Detection of Trace Explosives Using a Novel Sample Introduction and Ionization Method

**DOI:** 10.3390/molecules27144551

**Published:** 2022-07-17

**Authors:** Lingfeng Li, Tianyi Zhang, Wei Ge, Xingli He, Yunjing Zhang, Xiaozhi Wang, Peng Li

**Affiliations:** 1School of Electronic and Information Engineering, Soochow University, Suzhou 215006, China; lingfengli@suda.edu.cn (L.L.); tyzhangtyzhang1@stu.suda.edu.cn (T.Z.); geeewei99@163.com (W.G.); hexingli@suda.edu.cn (X.H.); yjzhang1223@suda.edu.cn (Y.Z.); 2College of Information Science and Electronic Engineering, Zhejiang University, Hangzhou 310027, China; xw224@zju.edu.cn

**Keywords:** trace explosives detection, dielectric barrier discharge, counter flow, mass spectrometry

## Abstract

A novel sample introduction and ionization method for trace explosives detection is proposed and investigated herein, taking into consideration real-world application requirements. A thermal desorption sampling method and dielectric barrier discharge ionization (DBDI) source, with air as the discharge gas, were developed. The counter flow method was adopted firstly into the DBDI source to remove the interference of ozone and other reactive nitrogen oxides. A separated reaction region with an ion guiding electric field was developed for ionization of the sample molecules. Coupled with a homemade miniature digital linear ion trap mass spectrometer, this compact and robust design, with further optimization, has the advantages of soft ionization, a low detection limit, is free of reagent and consumable gas, and is an easy sample introduction. A range of common nitro-based explosives including TNT, 2,4-DNT, NG, RDX, PETN, and HMX has been studied. A linear response in the range of two orders of magnitude with a limit of detection (LOD) of 0.01 ng for TNT has been demonstrated. Application to the detection of real explosives and simulated mixed samples has also been explored. The work paves the path to developing next generation mass spectrometry (MS) based explosive trace detectors (ETDs).

## 1. Introduction

The detection of explosives is of great importance for transportation security and public safety and, hence, has been subjected to intensive research study and commercial exploitation [1,2,3]. Among various application scenarios, quick detection of trace explosives in the field has drawn particular attention for both wide applicability and technical challenges. In most countries, authorities for air transportation have implemented mandatory requirements for the use of qualified explosive trace detectors (ETDs) to screen passengers and their luggage. However, the vast majority of current ETD devices are based on ion mobility spectrometry (IMS) [4,5], an analytical method detecting ionized molecules under atmospheric pressure, which has inherent methodological shortcomings of poor resolution, leading to a high false alarm rate.

It is well known that mass spectrometry, with the advantages of high sensitivity and specificity, is well suited to overcome those problems, and hence, much research work has been carried out in this particular field of trace explosive detection using MS based methods [6,7]. To detect trace explosives on a fast basis (seconds) in the field, some of the laboratory based standard configurations of MS technology have been eliminated, such as time-consuming sample pretreatment, the use of gas chromatography for front end separation, and large vacuum systems. A second consideration is to avoid fragmentation of the explosive molecules during the ionization process. The real-world sample will be a mixture, with both targeted explosive molecules and many others of unknown ingredients. If the analytes are critically fragmented during ionization (hard ionization), the chemical information of the analyte’s molecular weight will be lost, and it will be likely that the targeted analytes cannot be differentiated from the complex background matrix.

The development of ambient ionization techniques has promoted the application of mass spectrometry in the field of on-site rapid explosives detection [8]. Plenty of ambient ionization sources have been proposed for direct sampling and ionization of explosives under ambient conditions, such as desorption electrospray ionization (DESI) [9,10] and direct analysis in real time (DART) [11,12], low temperature plasma (LTP) [13,14,15,16] or dielectric barrier discharge ionization (DBDI) [17,18,19,20], atmospheric pressure chemical ionization (APCI) [21,22], photoionization (PI) [23,24], paper spray ionization (PSI) [25,26], and so on. However, most of these methods normally involve the use of expensive and specialized consumables such as solvent, dopant, and inert gas, which should be avoided in field applications. On the other hand, the ionization of the explosive compounds is highly dependent on the reactant ions and the ionization source employed with complex gas-phase ion chemistries.

In this work, a novel sample introduction and ionization method for trace explosives detection, based on the DBDI method with air as the discharge gas, was developed. The most commonly used method for the sample introduction of ETD is membrane based thermal desorption, which can effectively desorb explosives and inhibit interference of other non-volatile compounds at the same time. DBDI, which was first proposed by Zhang’s group [27], has demonstrated superior performance in the detection of explosives, including high sensitivity and low power consumption. However, most of the studies carried out with DBDI require inert gas for discharge, and the few attempts with air resulted in poor ionization efficiency and complex fragment ion composition, with NO_3_^−^ as majority. In this work, the reaction region was separated spatially from the discharge region to overcome the problem of unstable ionization caused by the direct contact of the analyte with discharge electrodes. The use of counter flow in the discharge region effectively removed the interference of ozone and other reactive nitrogen oxides. The reactive ions of NO_3_^−^ and [HNO_3_(NO_3_)]^−^ with high electron affinity, which are commonly found in discharge-based ion sources and seriously affect the ionization of explosive compounds, were greatly inhibited.

Therefore, filtered air, instead of inert gas, can be used for discharge in such a design. Detailed consideration and discussion of instrument design are described in the next section. The developed assembly of the sampler and ion source was coupled with a homemade portable digital linear ion trap mass spectrometer for characterization and optimization of the design parameters. Six representative explosive samples were tested for both counter and forward flow settings. The performance of such a system was explored under optimized conditions, with real-world samples and a mixture of explosives tested. Consumable free and soft ionization of explosive molecules with good sensitivity was realized. Through the research of this paper, a DBDI source with the advantages of long lifetime, large discharge area, and high reaction ion current was introduced into a portable mass spectrometer and applied to the field of rapid detection of trace explosives. It paved the ionization method’s path to develop next generation MS-based ETD. The abbreviations throughout this paper are summarized in Table 1 for reference.

## 2. Results and Discussion

### 2.1. Effect of Flow Field Configuration

The effect of the flow field at the region where the discharge occurs was examined first, as this is considered to be a major factor in the case of using air as a discharge gas. Mass spectra of the blank were measured first, as shown in Figure 1. It can be observed that with a counter flow of 800 mL/min, the reaction ions of NO_3_^−^ (*m*/*z* 62) and [HNO_3_(NO_3_)]^−^ (*m*/*z* 125) are nearly removed, presumably due to the removal of the neutral intermediate byproduct, such as NO_x_ and O_3_ [28,29,30]. Some background peaks of the ionization and reaction regions with relative low intensity can be found in the spectrum, while the reaction ions are not observed. It is probable that the reaction ion is O_2_^−^ (*m*/*z* 32), the same as the report of the corona discharge with the counter flow [21], beyond the mass range of the ion trap. Meanwhile, NO_3_^−^ and [HNO_3_(NO_3_)]^−^, with high intensity, are found with the discharge gas flow in the forward direction. The fact that few background peaks were obtained indicates that the NO_3_^−^ and [HNO_3_(NO_3_)]^−^ are not good choices as reaction ions, owing to their high electron affinity.

The position of the DBD discharge tube is also an important factor that affects both the flow field and the electric field experienced by the reactant ions, and hence was examined by 1 ng TNT with the sum intensity of [TNT–H]^−^ (*m*/*z* 226) and TNT^−^ (*m*/*z* 227), as shown in Figure 2 (counter flow rate: 800 mL/min; sample flow rate: 200 mL/min). As the DBDI source approaches the reaction chamber, the signal intensity increases and maintains at a high level, within a range of about 2 mm. This can be explained by the increased amount of reactant ions drawn by the electrical field as the DBDI source nears the reaction chamber. While the DBDI source further intrudes into the reaction chamber, the signal intensity declines rapidly. It is possibly due to the mixture of the discharge flow and sample flow, as a result of lower ionization efficiency. Therefore, in the following measurements, the exit of the DBDI tube was kept aligned with the first electrode of the reaction region.

### 2.2. Optimization of the Flow Rate

The effect of the flow rate of both discharge counter flow and sample flow was investigated after the configuration of the flow field was determined. Figure 3a gives the relation between the signal intensity of 1 ng TNT, [HNO_3_(NO_3_)]^−^, and NO_3_^−^ against variation in the counter flow rate through the discharge region. The intensity of [HNO_3_(NO_3_)]^−^ and NO_3_^−^ decreases substantially as the counter flow increases, meaning undesired neutral byproducts can be effectively removed with counter flow gas of certain speed. The relationship between the decrease in [HNO_3_(NO_3_)]^−^ and NO_3_^−^ and increase in ionization efficiency is also verified, which is consistent with the previous report [21]. In order to fully remove the neutral byproducts, the counter flow rate was set to 800 mL/min in all following measurements.

The effect of the sample flow rate was also examined, as shown in Figure 3b. As the sample flow rate increases, the initial increase in TNT signal intensity can be explained, as a larger flow can collect more TNT molecules from the thermal desorption sampler. The amount of analyte entering the reaction chamber will reach a maximum, as it will eventually be limited by the amount of sample introduced into the thermal sampler; further increase in sample flow leads to lower volume concentration and shorter retention time in the reaction chamber, resulting in a decrease in the measured signal. Therefore, in other experiments, the sample flow rates were all set to 200 mL/min.

### 2.3. Optimization of Discharge Conditions

Sinusoidal voltage of different amplitudes was applied to both the inner and outer electrodes to examine the effect of discharge polarity on ionization and to uncover the optimum discharge voltage, as shown in Figure 4. A signal intensity of 1 ng TNT was taken as the measure of ionization efficiency. The results show that the ionization efficiency is much higher, while the high voltage (HV) is applied on the inner electrode, which is consistent with the work of S. Brandt, et al. [31]. The substantial increase in the measured signal suggests that the direct contact of the HV electrode with the discharge gas can facilitate the generation of reactant ions. Moreover, the alternating voltage of the inner electrode offers a potential ingredient towards the reaction chamber, aiding in the transport of the reactant towards the reaction chamber.

From the observed ion products, both electrode settings had a similar effect on soft ionization. Taking TNT as an example, in either case, the deprotonated molecular ion [M-H]^−^ (*m*/*z* 226) and M^−^ (*m*/*z* 227) dominated on the measured mass spectra, with minimal other fragment ions or adduct ions observed. The observed result eliminated our initial concern for the effect of soft ionization while applying HV on the inner electrode. We ascribed this to the spatial separation of the reaction chamber from the discharge region and the introduction of a counter flow design.

For discharge voltage, it did not make much difference in our experiments when the peak-to-peak value of the applied sinusoidal voltage exceeded 4 kV. Therefore, 4 kV was chosen as the discharge voltage for all other measurements reported in this work, with 17 kHz as the frequency.

### 2.4. Soft Ionization of Explosive Molecules

After the optimization of basic operating parameters, six typical explosive samples were tested by the system, including TNT, 2,4-DNT, NG, RDX, PETN, and HMX, to examine the ionization and determine the characteristic ion product. The molecular structure and measured mass spectra, with forward flow and counter flow as comparison, are shown in Figure 5 and Figure 6.

For all six explosives, soft ionization was successfully achieved with the given conditions (see experiment section), resulting either in a deprotonated molecular ion ([TNT–H]^−^ (*m*/*z* 226) for TNT, [DNT–H]^−^ (*m*/*z* 181) for 2,4-DNT), molecular ion (TNT^−^ (*m*/*z* 227) for TNT), or adduct ion (mainly adduct with NO_2_^−^ or NO_3_^−^ for NG, PETN, RDX, and HMX). It is worth noting that the effect of counter flow on both soft ionization and ionization efficiency is not the same for different molecules. In the case of TNT and 2,4-DNT, the molecular ion totally disappeared from the measured mass spectra with the introduction of forward flow. The intensity of NG with forward flow is much lower than it is with counter flow, even with a relatively high concentration of 100 ng/μL. This is possibly due to the competition of the ionization process. For PETN and HMX, the existence of [HNO_3_(NO_3_)]^−^ did not seem to hinder the formation of adduct ions of [PETN + NO_3_]^−^ and [HMX + NO_3_]^−^, and the benefit of counter flow was higher ionization efficiency, as demonstrated by increased signal intensity. Finally, for RDX, the dominant adduct ion changed from [RDX + NO_3_]^−^ with forward flow to [RDX + NO_2_]^−^ with counter flow at a similar signal intensity. These results show that ionization of explosive molecules is highly dependent on reactive ions. Through eliminating neural discharge by-products such as ozone and nitrogen oxides, the formation of [HNO_3_(NO_3_)]^−^ and NO_3_^−^ can be largely inhibited, which greatly facilitates the ionization of explosive molecules. Although the fundamentals responsible for such a difference exceed the scope of this work and need to be further investigated by dedicated studies, this observation does suggest that, as the number of interested analytes increases, the optimized working conditions of the system may need to be further fine-tuned.

### 2.5. Analytical Performance

The overall performance of the system was evaluated in terms of sample analysis from a mixture, dynamic range, and limit of detection. In Figure 7, the ability of fast detection and discrimination of explosives from a mixture is demonstrated in four typical cases, which are the real explosive samples of Amatol and C-4, a simulated interference sample of TNT mixed with hand cream, and a mixture sample of TNT, 2,4-DNT, RDX, and PETN. Characteristic ion peaks for targeted explosives can be easily identified in all cases, suggesting the feasibility of applying the system in real sample analysis for field applications.

Although, in the application of ETD, quantification of the analyte is not the emphasis, it is still valuable to know the dynamic range of the instrument. With TNT as an example, the system was tested with various sample amounts and exhibited good linear response in the range of two orders of magnitude (0.01 to 1 ng/μL), as shown in Figure 8. In addition to desorption from the substrate, permeation through the membrane, transportation to the reaction chamber, ionization, and penetration of the sample orifice are consistent within a wide calibration range. The limit of detection for all six explosives was evaluated and summarized in Table 2. Experimentally determined LOD ranged from 0.01 ng for TNT to 0.5 ng for HMX, defined as a signal-to-noise ratio of 3:1. Considering the fact that we used an in-house constructed simple mass spectrometer with a discontinuous sample interface, which takes only a small fraction of total product ions, we expect the LOD can be further improved by coupling this sample introduction and ionization device to a better mass detector.

## 3. Considerations for Instrument Design

As briefly discussed in the introduction, to develop a technology platform which can be the basis for the next generation of ETDs, it is desired to overcome the shortcomings while maintaining the advantages of existing IMS based instruments. Current commercial ETD products usually use simple material such as Polytetrafluoroethylene (PTFE) or Nomax paper to transfer the analyte from the surface of suspected objects into the instrument, following thermal desorption of the analyte molecules from the sampling material. Targeted molecules are then ionized by either a radioactive ionization source such as Ni^63^ or by non-radioactive methods such as corona discharge (CD). The identity and quantity of ionized explosive molecules will be determined by their mobility and signal amplitude on the ion detector. LOD is normally around ~ng, as required by the standards issued by regulation authorities. The advantages include compactness (3–5 kg for the handheld model), ease-of-use, freedom from special consumables, and quick detection time (~s). The main limitation of such a product is its poor resolution (~30), which leads to a high false alarm rate and, hence, affects the throughput of the checkpoints.

The use of a mass spectrometer offers much higher resolution compared with IMS, at the cost of more complicated hardware and lower utilization of the analyte. However, due to the limits on detection time and possible complex sample composition, ionization and sample introduction methods have to be carefully considered. In this work, a thermal desorption sampler with Nomax paper as the analyte transfer substrate was adopted with a polydimethylsiloxane (PDMS) membrane to separate the desorption region. The advantages are two-fold. Firstly, the use of the transfer substrate allows sampling of a wide range of objects, including those which are hard to reach by the instrument itself. Secondly, the semi-permeable membrane blocks the interferences of dust, fiber, and airborne particles while allowing mostly explosive molecules to pass through.

For an ionization source, the use of a radioactive material is ruled out, first due to regulatory cost, and so is corona discharge for its poor durability (sharp discharge electrodes need to be replaced periodically). A DBDI source can theoretically address those problems. Instead of relying on the discharge of a fine tip where the maximum electrical field exists, as in the case of CD, with DBDI, the discharge happens between the surface of one electrode and the dielectric layer; the durability of the electrode and the distribution of the reactant ions are, hence, expected to be much improved. However, with the majority of studies on DBDI, special discharge gases such as helium and argon were used for better ionization effect. This may help to elucidate the ionization mechanism, but clearly will put restrictions on possible applications, especially with field deployable instruments. In a few cases, air was used instead for discharge [32,33,34], resulting in poor ionization efficiency and ion fragmentation due to the complex composition of plasma and, therefore, complicated ionization process. Such problems should be avoided from an instrument design perspective.

Counter flow was introduced by Y. Takada et al. [21] in the case of corona discharge ionization. In his design, an air flow carrying analytes travels in the direction away from the sample inlet of the mass spectrometer to blow away neutral by-products during the ionization process, such as O_3_ and NO_x_, and hence increase the ionization efficiency. The flow rate was found to be a crucial factor, leading to different ionization products [21,28]. This is of particular importance in the case of explosives detection, since nitrogen associated reactant ions generated by the discharge of air are known to play an important role in the ionization of common explosive molecules [21,28].

If the sample analytes directly overlap in the discharge plasma of both CD and DBD, the phenomenon of ion fragmentation and complex ionization products will occur. It is possibly due to the high density and energy distribution of free electrons inside the plasma. Moreover, different kinds of samples, especially high concentration samples, may cause discharge instability. It was therefore suggested to separate the discharge region from the ionization of analytes in space to achieve a better soft ionization effect, usually at the cost of trading off overall efficiency.

## 4. Materials and Methods

### 4.1. Materials and Reagents

HPLC grade acetonitrile was purchased from Tedia Co., Ltd. (Fairfield, OH, USA). Standard explosive samples of 2,4-DNT, TNT, RDX, PETN, HMX, and NG at a concentration of 1000 ng/μL were purchased from Beijing Shiji ’aoke Biotechnology Co., Ltd. (Beijing, China). A molecule sieve 13X was purchased from Sinopharm Chemical Reagent Co., Ltd. (Shanghai, China). Two real-world samples, including Amatol and C-4, were provided by a local government department. Hand cream, as a common interference sample, was purchased from the local supermarket.

### 4.2. Sample Preparation and Introduction

The working solutions of explosives were prepared by diluting the stock solution with acetonitrile to the target concentration. TNT, one of the most common explosives, was used to optimize the parameters of the platform with the concentration of 1 ng/μL. A standard liquid solution of analytes was added drop-wise onto the Nomex substrate using a pipette with 1 μL aliquot. After evaporation of the solvent in about 10 s, the substrate carrying the analyte was inserted directly into the desorption sampler. For the real-world samples, the surface of the sample was scrubbed with the Nomex paper to collect the analyte, followed by direct insertion of the substrate into the sampler. For the simulated interference test, the sample preparation process involved applying hand cream on the hands first, then wiping the hands with Nomex substrate, and then dropping 1 μL of TNT solution with the concentration of 1 ng/μL on the substrate. After evaporation of the solvent for about 10 s, the substrate carrying the analyte was inserted directly into the desorption sampler. For a mixed sample, 1 μL TNT, 2,4-DNT, RDX, and PETN solution with a concentration of 100 ng/μL was dropped onto the Nomex substrate, respectively. After evaporation of the solvent, the substrate carrying the analyte was inserted directly into the desorption sampler.

### 4.3. Design of the Platform System

Based on the aforementioned considerations, we have devised our current ionization and sampling design, as shown in Figure 9a. DBDI is employed with air as a discharge gas and sample carrier gas. Discharge gas flow and sample flow are both made of air filtered by the molecule sieve 13X and independently controlled for optimization. A separate reaction chamber is arranged, with the electrical field established and aligned with the direction of sample flow for negatively charged ions, to facilitate the reaction between reactive ions and neutral sample molecules. It is, therefore, expected that at the entrance point of mass spectrometer, the ionization of analyte molecules is maximized, and yet, without complicated product ions, since the analyte is not in direct contact with DBD plasma. The performance of such a system was evaluated by coupling to our in-house constructed miniature digital linear ion trap (DLIT) mass spectrometer. The schematic diagram of the whole platform system is shown in Figure 9b.

The sampling chamber was made of aluminum and heated up to 200 °C by a built-in ceramic heating plate. A semi-permeable PDMS membrane was used to separate the sampling chamber and ionization region. Evaporated analytes were carried away to the ionization region on the other side of the membrane by filtered air gas flow.

The DBD ion source comprised an inner electrode of a stainless-steel rod (diameter of 0.5 mm, length of 73 mm), dielectric of a quartz glass tube (outer diameter of 2.9 mm, thickness of 0.5 mm and length of 45 mm), and outer discharge electrode of copper belt (thinness of 0.15 mm, length of 30 mm). All parts were fixed through a tee-junction to two fixtures made of polyether ether ketone (PEEK), to ensure that the inner electrode and dielectric tube would be coaxial. The inner electrode and outer electrode were placed at the position of the quartz glass tube outlet with a distance of 2 mm. A sinusoidal voltage of 5 kV at a frequency of 17 kHz, produced by a signal generator (CTP-2000K, Najing Suman Plasma Co., Ltd., China), was applied to the electrode via a high voltage capacitor (C1) of 1 nF for the generation of atmospheric pressure plasma. Meanwhile, the DC potential of the discharge electrodes was −400 V, which is lower than the first electrode of the reaction region, with 150 V to pull the negative reaction ions into the reaction region. To protect the DC source module, the resistors (R1) of 20 MΩ were connected to the inner and outer electrode, respectively, as shown in Figure 9a.

The reaction chamber between the DBD ion source and mass spectrometer was made of PEEK and heated to 200 °C. Six metal electrodes were inserted to form a uniform electric field of 133 V/cm in the reaction region. Air was used for both discharge of DBD and of carrier gas for the analyte inlet, with the discharge gas flow ranging from 200 mL/min to 1200 mL/min and the sample gas flow ranging from 50 mL/min to 300 mL/min. The whole ionization and sample introduction setup functioned under atmospheric pressure.

The sample introduction and ionization assembly was attached to a homemade portable digital linear ion trap mass spectrometer with the original ion source removed, which has been previously described in detail [35]. Briefly, the instrument uses a linear ion trap with a digital waveform with a low voltage (±100 V) as the mass detector, and a discontinuous atmospheric pressure interface (DAPI), as described by L. Gao et al. [36], covered a mass range of 50–500 Da with a unit resolution at a scan rate of 10,000 Da/s.

## 5. Conclusions

In this study, we have developed a sample introduction and ionization method for the detection of trace explosives based on a step-by-step discussion of the requirements for a field deployable instrument. A device based on thermal desorption and DBDI with air as the discharge gas was constructed to implement the method, and working conditions were examined and optimized. By coupling this device to a homemade mass spectrometer, soft ionization of six common explosives was successfully achieved, with distinctive product ion identified. The analytical performance of the system was evaluated, and LOD was found to be improved over that previously reported for air-based DBDI. Real explosives and simulated mixed samples could be well detected, indicating the feasibility of applying the system in real sample analysis for field applications. In future work, we will further develop the compact gas supply system and power supplier of the sinusoidal high voltage for the DBDI source to miniaturize the whole system, which can be used portably and lightly in the security checkpoints. The result of the work laid a solid foundation for the development of MS-based ETD instruments, one which can overcome the limitations of current solutions, without using consumables including solvent and inert gas, and has potential in a wide range of applications.

## Figures and Tables

**Figure 1 molecules-27-04551-f001:**
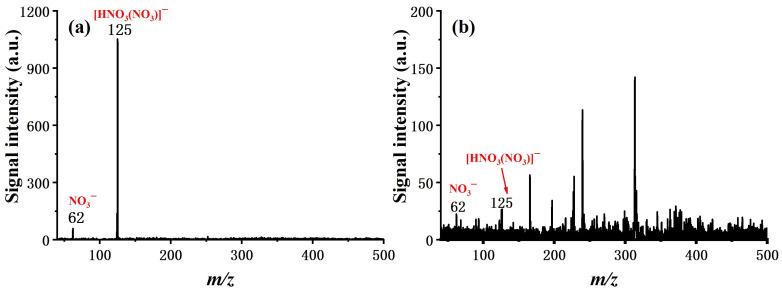
Mass spectra of background in the case of discharge gas flow (**a**) in forward direction and (**b**) in counter direction. The sample flow was set to 0 mL/min in either case.

**Figure 2 molecules-27-04551-f002:**
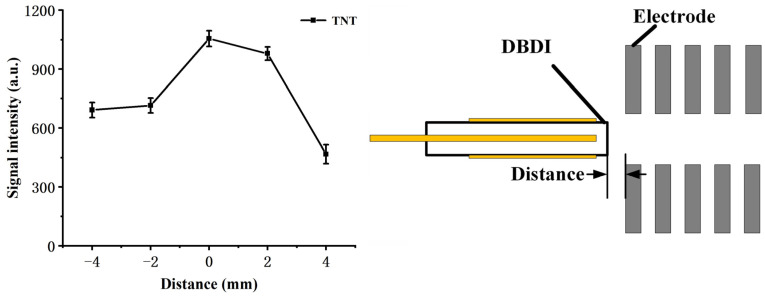
Effect of the distance between the DBDI source and reaction chamber, measured by a signal intensity of 1 ng TNT.

**Figure 3 molecules-27-04551-f003:**
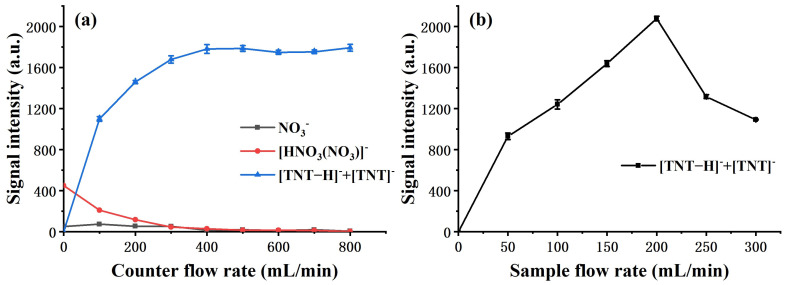
Effect of the flow rate (**a**) signal intensity of 1 ng TNT, [HNO_3_(NO_3_)]^−^, and NO_3_^−^ versus counter flow rate; (**b**) signal intensity of 1 ng TNT versus sample flow rate.

**Figure 4 molecules-27-04551-f004:**
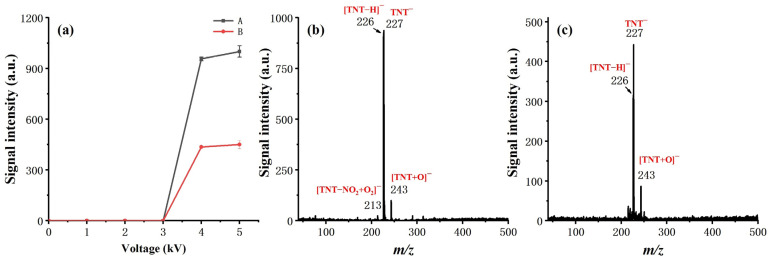
Effect of discharge polarity and voltage, measured by a signal intensity of 1 ng TNT. (**a**) Curve A: HV is applied on the inner electrode; curve B: HV is applied on the outer electrode; (**b**) Mass spectrum of 1 ng TNT, to which HV is applied on the inner electrode; (**c**) Mass spectrum of 1 ng TNT, to which HV is applied on the outer electrode.

**Figure 5 molecules-27-04551-f005:**
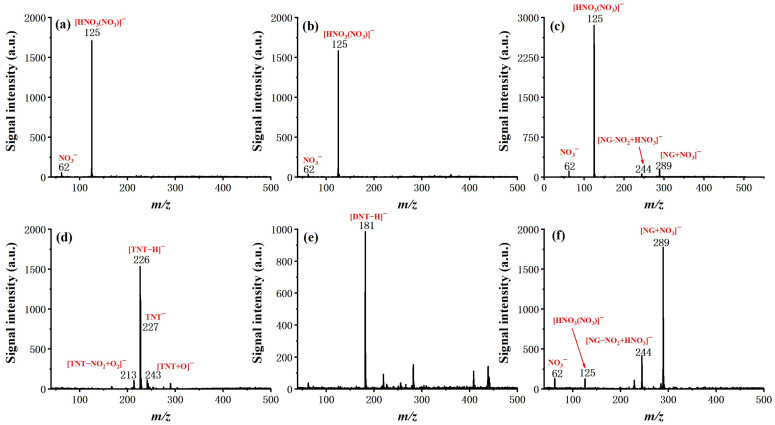
Mass spectra of TNT, 2,4-DNT, and NG with 100 ng, respectively, with discharge gas flow of 800 mL/min. (**a**) TNT with forward flow; (**b**) 2,4-DNT with forward flow; (**c**) NG with forward flow; (**d**) TNT with counter flow; (**e**) 2,4-DNT with counter flow; (**f**) NG with counter flow.

**Figure 6 molecules-27-04551-f006:**
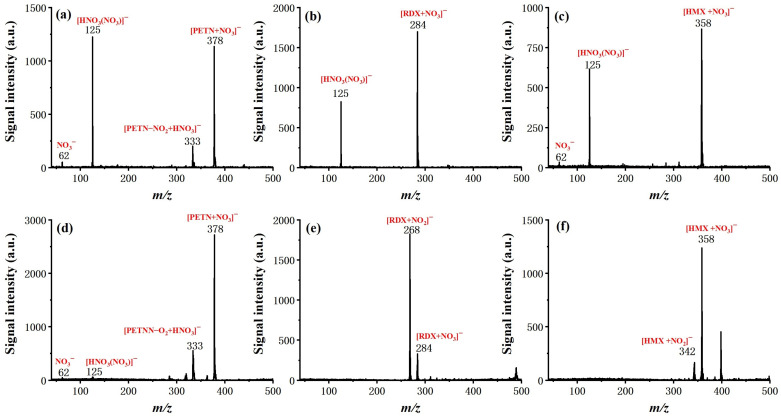
Mass spectra of PETN, RDX, and HMX with 100 ng, respectively, with discharge gas flow of 800 mL/min. (**a**) PETN with forward flow; (**b**) RDX with forward flow; (**c**) HMX with forward flow; (**d**) PETN with counter flow; (**e**) RDX with counter flow; (**f**) HMX with counter flow.

**Figure 7 molecules-27-04551-f007:**
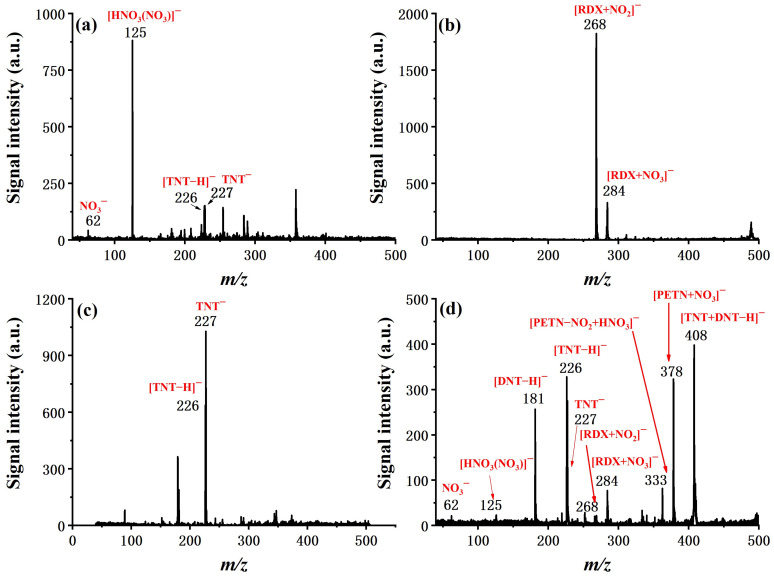
Detection and identification of explosives in mixture sample of (**a**) Amatol (AN, TNT, and wood powder), (**b**) C-4 (RDX, plastic binder, plasticizer, and taggant), (**c**) 1 ng TNT mixed with hand cream, and (**d**) mixture of TNT, 2,4-DNT, RDX, and PETN with an amount of 100 ng separately.

**Figure 8 molecules-27-04551-f008:**
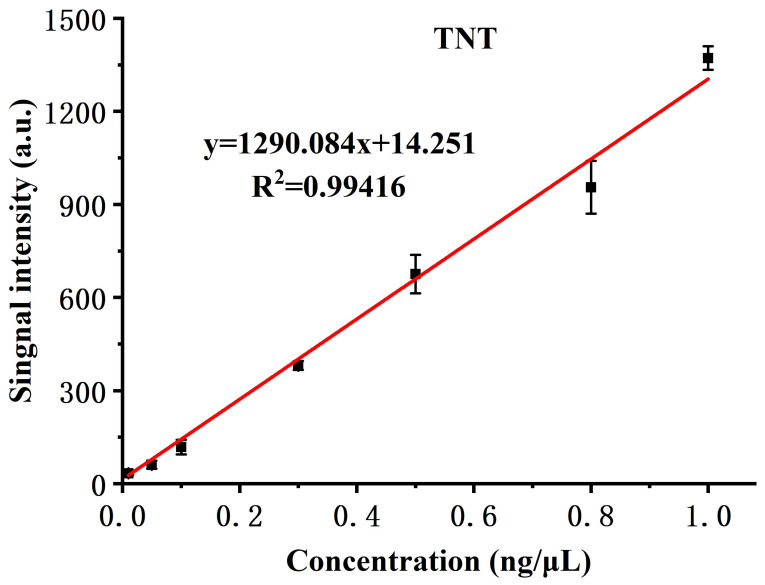
Dynamic range and linearity, in case of TNT.

**Figure 9 molecules-27-04551-f009:**
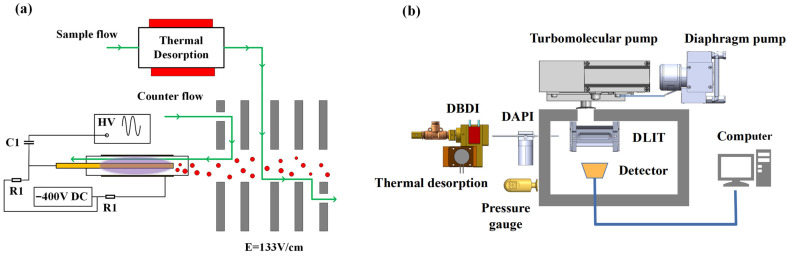
The schematic diagram of the homemade mass spectrometer. (**a**) The schematic design of the ionization and sampling design; (**b**) the schematic diagram of the platform system.

**Table 1 molecules-27-04551-t001:** The full names of the abbreviations or symbols in this study.

Abbreviations or Symbols	Full Names
DBDI	dielectric barrier discharge ionization
LOD	limit of detection
MS	mass spectrometry
ETD	explosive trace detector
IMS	ion mobility spectrometry
PTFE	polytetrafluoroethylene
CD	corona discharge
PDMS	polydimethylsiloxane
DLIT	digital linear ion trap
PEEK	polyether ether ketone
DAPI	discontinuous atmospheric pressure interface
HV	high voltage
2,4-DNT	2,4-Dinitrotoluene
TNT	Trinitrotoluene
RDX	1,3,5-Trinitroperhydro-1,3,5-triazine
PETN	pentaerythritol tetranitrate
HMX	1,3,5,7-Tetranitro-1,3,5,7-tetrazocane
NG	nitroglycerin

**Table 2 molecules-27-04551-t002:** Characteristic product ion and LOD of six tested explosives.

Sample	Ions Observed	Major Ion	LOD (ng)
2,4-DNT	181	[DNT–H]^−^	0.01
TNT	213	[TNT–NO_2_ + O_2_]^−^	0.01
	226	[TNT–H]^−^	
	227	[TNT]^−^	
	243	[TNT + O]^−^	
RDX	268	[RDX + NO_2_]^−^	0.05
	284	[RDX + NO_3_]^−^	
HMX	342	[HMX + NO_2_]^−^	0.5
	358	[HMX + NO_3_]^−^	
PETN	333	[ PETN –NO_2_ + HNO_3_]^−^	0.1
	378	[PETN + NO_3_]^−^	
NG	244	[NG–NO_2_ + HNO_3_]^−^	0.1
	289	[NG + NO_3_]^−^	

## Data Availability

The data presented in this work are available in the article.

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
