# Peer review of "Detection of Trace Explosives Using a Novel Sample Introduction and Ionization Method"

_molecules, 2022, doi:10.3390/molecules27144551_

Round 1
Reviewer 1 Report
Accelerating services and increasing safety at the entrance gates of sensitive transportation items such as airports, trains, and subways using new technical ideas can be valuable. This research has been designed and conducted in this regard. It seems after major revision it can be considered for publication. Here are a few of these revisions needed, for example.
The scientific English language of the article must be revised.
e.g.,
‘The decrease of the signal while the DBDI source goes further is possibly due to the disturbance of flow field, resulting more mixing of discharge flow and sample flow and lowered ionization efficiency.’
‘It features thermal desorption, dielectric barrier discharge ionization (DBDI) source, air as discharge gas, counter flow introduction, and separated reaction region with ion guiding electric field.’
‘Studies on soft ionization in both CD and DBDI also revealed direct overlapping of discharge plasma and sample analyte results in ion fragmentation and complex ionization product, possibly due to high density and energy distribution of free electron inside the plasma.’
….
Sentences are ambiguous and hard to flow to some extent.
Please define the abbreviation’s full names at a table and summarize the abbreviations and symbols.
At the end of the introduction section, the purpose of this research and the specific benefits you are looking for in this research should be clearly explained. Please consider.
The calibration method of the new instrumental device and set-up of the proposed new analysis method should be clearly explained.
For example, what mass spectra method did you use? NMR, RAMAN, and the like.
Or, if these measurements and analyses were measured directly by the new set-up (figs 5 to 8), what basis did you have for the accuracy and precision of the measurement? Please explain.
Can this device be used portably and lightly in the input gates? Or is there a need to build a fixed location like x-ray strips? Please add comments on this.
Reviewer 2 Report
The present manuscript entitled "Detection of trace explosives using a novel sample introduction and ionization method" by Lingfeng Li, Tianyi Zhang, Wei Ge, Xingli He, Yunjing Zhang, Xiaozhi Wang, Peng Li (molecules-1806592) is written correctly and has a good structure; moreover, it has all the necessary parts. The article is very interesting from an analytical point of view; therefore, it should interest the reader. I only have a few questions about the article. The paper meets Molecules' requirements, and I recommend the article for publication in Molecules following the common editing stage. My current decision is a minor revision.
More specific comments and observations are presented below.
1. The authors presented a simulation of interference tests. What can be done in the event of strong interference effects? How would you deal with them? What types of interference effects could occur?
2. Figure 1. Please add an explanation of abbreviations in the description of the figure. There is an error (figure 1b) in the word "detector".
3. Was it planned to apply the experimental planning methods to the optimization part?
4. “relationship” is mentioned. This term should be changed to "relation". The relationship tends to be used more broadly to describe the interactions between specific people or smaller groups of people.
5. Table 1. Shouldn't there be an ng/µL unit for LOD? In this table, you can also include parameters such as LOQ, linearity range, R2, intercept, and slope.
6. Does the developed method have disadvantages? What are the limitations?
7. Conclusion. Please, emphasize clearly the advantages of the research carried out.
8. It would be worthwhile to evaluate the method using RGB Additive Color Model to Analytical Method Evaluation or AGREE-Analytical GREEnness Metric Approach.
I hope that the comments presented will help improve the article.
